# Quantitative Self-Assessment of Exposure to Solvents among Formal and Informal Nail Technicians in Johannesburg, South Africa

**DOI:** 10.3390/ijerph20085459

**Published:** 2023-04-11

**Authors:** Derk Brouwer, Goitsemang Keretetse, Gill Nelson

**Affiliations:** School of Public Health, Faculty of Health Sciences, University of the Witwatersrand, Johannesburg 2193, South Africa; derk.brouwer@wits.ac.za (D.B.); gill.nelson@wits.ac.za (G.N.)

**Keywords:** passive sampling, VOCs, participatory research

## Abstract

Participatory research, including self-assessment of exposure (SAE), can engage study participants and reduce costs. The objective of this study was to investigate the feasibility and reliability of a SAE regime among nail technicians. The study was nested in a larger study, which included exposure assessment supervised by experts, i.e., controlled assessment of exposure (CAE). In the SAE approach, ten formal and ten informal nail technicians were verbally instructed to use a passive sampler and complete an activity sheet. Each participant conducted measurements on three consecutive days, whereafter the expert collected the passive samplers. Sixty samples were, thus, analyzed for twenty-one volatile organic compounds (VOCs). The reported concentrations of 11 VOCs were converted into total VOC (TVOC) concentrations, adjusted for their respective emission rates (adj TVOC) to allow comparison within and between nail technician categories (formal vs informal), as well as assessment regimes (SAE versus CAE), using the data from the main study. In total, 57 SAE and 58 CAE results were compared, using a linear mixed-effects model. There were variations in individual VOC concentrations, especially for the informal sector participants. The major contributors to the adj TVOC concentrations were acetone and 2-propanol for the formal category, whereas ethyl- and methyl methacrylate contributed most to the informal nail technicians’ total exposures. No significant differences in adj TVOC-concentrations were observed between the assessment regimes, but significantly higher exposures were recorded in the formal technicians. The results show that the SAE approach is feasible in the informal service sector and can extend an exposure dataset to enable reliable estimates for scenarios with substantial exposure variations.

## 1. Introduction

Citizen science, i.e., a voluntary collaboration among scientists and non-specialists to achieve both scientific and societal goals, is an emerging form of scientific inquiry that is growing in popularity in the environmental sciences [1,2]. Participatory knowledge generation is the main goal [1]. However, education and engagement of the public with scientific discoveries are other important objectives. Introducing citizen science into occupational health is considered a potentially innovative and economical approach, especially since resource constraints challenge this field [2]. From a measurement strategy perspective, daily monitoring of workers’ exposure is the optimum approach. However, currently, the selection of an appropriate measurement strategy is a compromise between statistical efficacy, i.e., minimizing the variance of an unbiased estimate of the target exposure variable, and associated costs [3]. Traditionally, in occupational exposure assessment, much attention is paid to generating high-quality data with high chemical analytical precision and accuracy, requiring highly skilled experts and high-precision devices. However, Nicas et al., (1991) have demonstrated that measurement error is often small relative to the environmental variability of concentrations in the workplace [4]. Therefore, efforts were taken to reduce sampling costs, for example, by employing passive, instead of active, sampling [5]. Another approach to reducing the cost of exposure sampling was introduced by Liljelind and co-authors in the early to mid-2000s [6,7,8], and this was later explored by others [9,10,11]. The concept of self-assessment of exposure (SAE) implies that workers assess their own exposures, instead of occupational hygienists. The basic idea was that an extension of the number of data points could be achieved without additional (researcher) costs. Quantitative self-assessment of inhalation exposure relied mainly on passive sampling of volatile organic compounds (VOCs) and gases. Hertsenberg et al., (2006) reported the results of a quantitative study on VOC exposure amongst shoe repair men, using two different regimes of self-assessment, i.e., different levels of expert supervision [9]. The group with the lowest level of supervision received only written instructions on the use of the passive samplers (diffusion badges), whereas participants in the other group received extensive oral instructions, followed by controlled assessment of exposure. Petterson et al., (2008) conducted a study among sawmill workers, aiming at the relationship between measurement frequency of terpenes, variation of concentrations using diffusive samplers, and workers’ propensity to perform exposure measurements [10]. Workers were instructed both in written form and orally. The number of sampling days was not prescribed, but it was decided by the individual worker. No supervised measurements were conducted in this study. Hedmer et al., (2017) applied self-assessment of exposure for tunnel construction workers [11]. Workers received written instructions on handling the passive samplers (diffusion tubes) to assess exposure to NO_2_. A second, randomly selected group of workers, performing the same tasks, was monitored under the supervision of an occupational hygienist. Although the reliability of SAE data varied between the studies, it was concluded that workers, if well instructed and motivated, can generate reliable exposure measurements. Silicone wristbands have recently been deployed as personal passive sampling devices to assess dermal and inhalation exposure to various VOCs, semi-volatile organic compounds (SVOCs), and pesticides [12]. The use of wristbands was also piloted in nail salons to sample SVOCs. However, they were not used a self-assessment tools [13].

To investigate the reliability of SAE, especially in the informal service sector, we report on the results of a self-assessment study in which nail technicians in both formal and informal nail salons measured inhalation exposure to VOCs using passive diffusion samplers. Therefore, we compared the central tendency of VOC exposure concentrations of a SAE nail technician group with a group of nail technicians for which exposures were measured under the conventional controlled assessment of exposure (CAE) regime.

## 2. Materials and Methods

### 2.1. Study Population 

This comparative study had a cross-sectional design and was nested in a larger study in nail salons in the City of Johannesburg [14], hereafter referred to as the controlled assessment of exposure (CAE) study. Convenience sampling was used to select ten nail salons, representing the informal sector, as well as six salons, representing the formal sector, assuming that 30 samples for each sector and exposure assessment regime would be sufficient to demonstrate differences for a hypothesized moderate effect size of 0.48. The formal nail salons were franchises of one of the largest local beauty therapy companies in South Africa, which permitted them to approach the salons in the northern suburbs of Johannesburg. The informal nail salons comprised unregistered nail salons in the central Johannesburg Braamfontein area, some of which operated inside hairdressing salons. Before the exposure assessment was conducted as part of the main study, salons were visited, the purpose of the study was explained, and the nail technicians were invited to participate. Nail technicians who also performed hair and other cosmetic treatments, and those involved in other activities outside work, which may generate the release of solvents, were excluded from the study.

All nail salons in this study offered essential nail services, including manicures and pedicures, as well as nail polish, gel, and acrylic applications. Most nail applications began with an acetone soak-off process to remove old nail applications. The most common applications in the formal and informal nail salons were gel and acrylic nail applications, respectively. All the formal nail salons used similar products from known brands, procured through a structured company system. In the informal nail salons, such a central procurement system was lacking. In formal salons, four to six nail technicians usually offered nail treatment services simultaneously; in informal salons, only a single technician, or two nail technicians, worked simultaneously with a hairdresser, in most cases. Further details on the nail technicians and the nail salons are reported elsewhere [14].

If a nail technician agreed to participate in the study, i.e., to self-assess their personal exposure on three consecutive workdays, oral instructions were given on how to open, attach, close, and store the passive samplers (diffusion tubes), as well as how to complete the worksheets. The worksheets facilitated the participant to record information, such as the time the passive sampler was opened and donned, the number and types of nail applications performed, and the duration of the nail applications. In most cases, the instructions were given on a Monday, and the sampling was conducted on Tuesday through Thursday afternoon, after which the samples were collected. Finally, the researcher checked and noted the sampling tubes’ status and the worksheets’ completeness. Where information was missing, attempts were made to retrieve this from the participants. 

Ten nail technicians from both formal and informal sectors were included in the self-assessment study, resulting in 30 measurements for each sector. Unfortunately, COVID-19 restrictions resulted in a time gap between the self-assessment and the controlled-assessment of the main study. The self-assessment of the informal sector was conducted in June 2021, whereas the other sampling campaigns were conducted in the last quarter of 2021. Consequently, it was difficult to recruit the same technicians from the main study (controlled assessment) into the self-assessment study due to business closures and dismissals. Only eight nail technicians, i.e., four from each sector, participated in both assessments. 

### 2.2. Exposure Assessment 

Personal exposure measurements were collected using diffusion/passive samplers (Radiello^®^ Passive sampler, Sigma Aldrich, Kempton Park, South Africa). Before sampling, the sorbent cartridge was removed from its glass storage tube and inserted into the diffusive body; whereafter, it was attached to the participant’s lapel. Sampling devices were deployed to measure exposures for at least 80% of the work shift over three consecutive days, for each nail technician. A total of 60 personal exposure measurements in the formal (*n* = 30) and informal (*n* = 30) nail salons were collected. Field blanks were also collected during the controlled assessment to check for contamination during transport. At the end of the sampling period, the participant detached the sampler from the lapel, placed it in the glass tube, capped and labelled it, and stored it at <5 °C until it could be analyzed. The same procedure was followed during the controlled assessment; however, the investigator attached and removed the sampler and recorded the nail applications, etc.

### 2.3. Chemical Analysis 

Details of the chemical analysis have been published elsewhere by Keretetse et al. [14]. In brief, after extraction of the sorbent tubes in CS_2_, gas chromatography with a flame ionization detector was used to quantify the concentrations of VOCs. Chromatographic separation was performed using a DB-624 column (0.25 mm ID, 30 m length, 1.4 μm film thickness) for quantification, as well as a DB-WAX column (0.25 mm ID, 30 m length, 1.5 μm film thickness) for verification, with a split injection of 3 μL and split flow of 20 mL/min. The carrier gas was nitrogen at a constant pressure of 5 psi. The injector temperature was 220 °C, and the detector temperature was 200 °C. The MSD ChemStation macro program extracted Peak areas, adjusted for internal standards, and transferred them to an Excel spreadsheet. The average concentration over the exposure period was calculated from the mass of the analyte found on the cartridge and the reported duration of sampling.

Based on a review of the literature and the information obtained from the labels of the products used, or the safety data sheets (SDS) or the ingredients list (if any), 21 VOCs were analyzed. The samples (batches) collected during the informal sector self-assessment were sent to the laboratory for analysis much earlier than the batches with other samples. Unfortunately, only 15 similar VOCs were common in both batches. To simplify the calculation, the analytical limit of detection (LoD) and limit of quantification (LoQ, 3.3 times LoD) for all compounds were set to 0.50 µg/mL and 1.65 µg/mL, respectively. 

### 2.4. Data Analysis 

Data pre-processing, including data cleaning and visualisation, was conducted using Microsoft Excel 2016. Descriptive statistics were calculated using ExpoStats, a Bayesian toolkit [15]. Imputation for data below LoQ was performed using Expostats—a NDexpo /RoS tool. NDexpo implements a rigorous censored data treatment method by labelled regression on order statistic [16]. For the data in the present study, the analytical LoQ was converted into a LoQ sampling concentration of 0.033 mg/m^3^. Since measured exposure concentrations appeared to be lognormally distributed, all statistical analyses were conducted on ln-transformed concentrations. Eleven VOCs, with a detection frequency of ≥30%, were selected from the 21 reported VOCs for further statistical analysis. 

Since both the number and the type of VOC varied within and between the sectors, the so-called adjusted total VOC concentration (TVOC) was calculated to enable comparison. The adjTVOC is defined as the total of the selected VOCs’ concentrations per subsector, each corrected by their respective evaporation rate relative to the lowest evaporation rate of the selected VOC, i.e., d-limonene (Appendix A Table A1). The evaporation rates were calculated using Hummel’s equation, which was incorporated into the IHMod^TM^ 2.0 mathematical model supporting files [17]. The equation requires only the molecular weight, vapour pressure, air velocity, pool size, ambient pressure, and liquid temperature, or estimates of these quantities, to approximate the overall evaporation rate in mass/time/unit area. The ratio of the evaporation rate of each VOC to that of d-limonene was used to correct the concentration of each individual VOC in what is referred to as the weighted concentration. The sum of these individual VOC-weighted concentrations was the adjusted TVOC concentration in mg/m^3^. 

JASP (JASP Team, version 0.16.3) [18] was used the estimate the effect of the exposure assessment regime, i.e., the self-assessment of exposure (SAE) and controlled-assessment of exposure (CAE), with a mixed linear effect model. In the model, the natural logarithm of the adjusted TVOC-concentration is the dependent variable, whereas the assessment regime (SAE or CAE), as well as the sector category (formal or informal), are the fixed effects. Both the participant and the day of measurements represent the random effects. The model was fitted using maximum likelihood, and random effects are assumed to be statistically independent. The model can be described as follows.
(1)$$Y_{ijrcd}=\beta_{0}+{\;\beta }_{r}+\beta_{c}+\alpha_{i}+\alpha_{j\;}+\varepsilon_{ijrc}$$
where *Y_ijrcd_* = the ln-transformed TVOC concentration of the *i*-th worker, measured on the *j*-th day, under the assessment regime *r* in the category *c*; β_0_ denotes the intercept, whereas β*_r_* and β*_c_* represent the fixed effects of the assessment regime and the category, respectively. The random effects of the worker and the sampling day are represented by α*_i_* and α(*_j_*), respectively. ε*_ijrc_* denotes the random error. The nonparametric Wilcoxon signed-rank test was used to test differences between the two assessment regimes, based on the GSDs for the same VOC.

## 3. Results

Nearly all formal nail technicians completed data capture sheets. If any data were missing, such as the number of clients or type of treatments, this information was retrieved from the client registration system when the researcher collected the forms and the samplers. In contrast, since such a system was lacking for the informal nail technicians, 15 data sheets were missing from five nail technicians.

Twenty-one VOCs were analyzed in total; however, only 15 VOCs from the SAE regime in the informal sector were analyzed. This included 1-butanol, which was not analyzed for all the other groups. Table 1 shows the analyzed VOCs and the samples with values >LoQ. 

From the 21 VOCs reported, 11 showed a detection frequency of >30% of the limit of detection (LoD) within the same sector. Methyl-isobutylene-ketone (MIBK) and trichloroethylene were not detected in any sample, whereas chloroform was detected only in the informal sector. Although benzene was detected in some samples, the concentrations were close to LoQ. Field blanks, which were collected and analyzed at each salon, showed negligible VOC levels, confirming that transport, storage, and handling activities did not contaminate the sampling badges. Based on the percentages detected, the 11 VOCs were selected for further analysis, i.e., acetone, n-butyl acetate, ethyl acetate, ethanol, ethyl methacrylate, d-limonene, methyl-methacrylate, 2-propanol, propyl acetate, toluene, and white spirits (Table 2). For the SAE in the informal sector, three results were excluded from further analysis; two resulted from a shorter-than-optimal duration of monitoring (<3 h), and one result showed concentrations of all VOCs below the LoD. During the CAE, 59 personal exposure measurements were collected, as one informal study participant was not present for work on the third assessment day. Since the results for one formal nail technician were all below LoD, this sample was excluded. Hence, 58 CAE- and 57 SAE-sampling results were further analysed. 

Table 2 shows that, for the formal sector, both the SAE and CAE regimes revealed that the VOCs with the highest concentrations were acetone and 2-propanol. The GM of the acetone concentrations was 65.7 and 108 mg/m^3^ for the SAE and CAE, respectively, with a much higher GSD (6.63) for the SAE compared to the CAE regimen (2.49). The GMs of the 2-propanol concentrations for the two assessment regimes were much closer, i.e., 46.7 versus 43.7, with a higher GSD for the SAE compared to the CAE, i.e., 2.21 versus 1.72. This demonstrates a general trend, wherein the GSDs of the VOC concentration distributions were slightly higher for the SAE regime, although they were not statistically significant (*p* = 0.064).

In the informal sector, acetone was also the VOC with the highest concentrations during both regimes. During the SAE regime, concentrations of ethyl-methacrylate and methyl-methacrylate showed the second-highest and third-highest GM, respectively, where the order was swopped during the CAE regime. The concentration distributions showed very large GSDs for both these VOCs. No statistically significant differences were observed in the GSD for all VOCs for both assessment regimes (*p* = 0.55).

The major contributors to the adjusted TVOC concentrations are depicted in Figure 1. For the formal sector, the adjusted TVOC was dominated by 2-propanol and acetone, whereas methacrylates were the main contributors in the informal sector. The picture for both regimes per sector is similar. The adjusted TVOC concentration distribution per sector and assessment regime is depicted in Figure 2. The mixed effects model estimates are shown in Table 3. Analyses were based on ln-transformed data, and the model results were back-transferred. The grand total of observations showed a significant difference in the restricted maximum estimates of the GMs for both sectors (*p* = 0.011); however, the effect of the assessment regime was not statistically significant (*p* = 0.37).

## 4. Discussion

Nail treatments require the use of various products, depending on the type of treatment. As was demonstrated by Keretetse et al. [14], the type of treatment demands different products. Therefore, a nail technician’s exposure to VOCs will heavily depend on the number of clients and treatments performed. Consequently, large exposure variances are expected, as demonstrated by substantial GSDs for most of the VOCs. In our previous paper, we showed that, due to bystander exposure, the variation in exposure levels of the nail technicians in the formal sector was smoothed [14]. In contrast, the informal technicians’ personal VOC exposure was determined entirely by their own activities. For example, the exposure to methacrylates is associated with artificial nail treatments, and Table 2 shows that, especially for the informal sector, the GSDs of the methacrylates’ concentrations were large. The variability of exposure to these VOCs during the SAE assessment regime was enhanced by ‘slow business’, immediately following the lifting of stringent COVID-19 restrictions. Unfortunately, the information on the number of clients and types of treatments was missing for half of the informal SAE participants. Soaking off residues from previous nail treatments is part of every treatment, with acetone being the major VOC. Remarkable is the high frequency and the relatively high concentration of 2-propanol in the formal sector. This VOC is also a constituent of hand sanitizer, an important part of the COVID-19 non-pharmaceutical intervention protocols to which strict adherence was observed in the formal sector. 

To enable comparison and to adjust for overweighting the results of the relatively higher volatile VOCs, we developed the concept of VOC–concentration, adjusted for the VOC-specific emission rate. Adding all adjusted VOC–concentrations resulted in an adjusted total VOC–concentration as a metric for cumulative exposure to the VOCs detected and included in our analysis. This differs from other approaches where TVOC was reported as the sum of the concentrations of compounds detected in the range of C_6_–C_16_ [19], or in the form of a VOC score [9]. Our approach, i.e., a surrogate d-limonene equivalent concentration, is quite comparable with the toluene-equivalent concentration, as deployed by Wallenius et al., (2022) [20].

This study aimed to investigate whether self-assessment of exposure could reliably estimate the true distribution for the nail technicians under investigation. Our results indicate that the different exposure assessment regimes did not significantly affect the exposure assessment results. Consequently, the results of both CAE and SAE could be pooled, revealing a significant difference in the adjusted TVOC concentrations between both categories. Whereas, based only on the CAE or SAE data, no statistically significant differences between the formal and informal nail technicians were observed. Our results, regarding the performance of SAE compared to CAE, are aligned with the results and conclusions of other quantitative studies of self-assessment exposure with similar designs.

In the study reported by Hertsenberg et al., (2006), the authors found no significant differences in the sum score for VOCs. However, the exposure variability appeared to be significantly higher for the group that received written instructions. The comparison of the CAE and SAE within the group that received face-to-face instructions showed no differences in either the GM or the exposure variability. 

The study by Hedmer et al., (2017) revealed that there were no statistically significant differences in the mean exposures observed between the controlled and the self assessment groups. The authors concluded that, by employing SAE, more workers could be included in the study, especially under the harsh work environment conditions of the tunnel construction workers and limited accessibility to the (tunnel) workplace for experts to conduct the measurements [11]. Since the study reported by Pettersson et al., (2008) had a different aim, i.e., to investigate whether the average and variation in exposure levels were of importance in the workers’ performance of exposure measurements, the authors reported the correlation between each worker’s number of measurements and their mean exposure levels and the coefficient of variance, respectively [10]. No statistically significant correlations were found. An interesting finding was that, in all four workplaces included in the study, the supervisor, contrary to the instructions, organized the measurements. Therefore, the authors concluded that a formal organizational structure is needed for systematic monitoring beyond letting the workers conduct a simple SAE. 

Several limitations could have affected the results of our study. Environmental conditions, e.g., temperature and relative humidity, were not recorded during the SAE measurements. Since all CAE measurements and the SAE measurements in the formal sector were performed on three consecutive days in the early summer period (October 2021), the average temperature ranged only from 23 to 28 °C [14]. As reported, the SAE for the informal sector took place during the winter period (June 2021). Hence, we expect somewhat lower average temperatures over the workday. However, it is unclear what the impact of temperature on the VOC concentrations might have been. On the one hand, lower temperatures will decrease volatilization, whereas, on the other hand, less ventilation by keeping the door closed might have increased the VOC concentrations. In our calculation of the TVOC concentration for the comparison between SAE and CAE, we considered only the VOCs detected in 30% or more samples for both regimes. Consequently, we excluded the results for ten VOCs, including benzene, chloroform, and xylene. This hampers interpretation according to health risk. It also affected the comparison between the formal and the informal sector, as ten and eight VOCs were included in the TVOC calculations for the formal and informal sectors, respectively. Importantly, the fact that 2-propanol was not analyzed in the informal sector during the SAE regime impacts the comparison between the two sectors, as 2-propanol contributes most to the adjusted TVOC concentration in the formal sector. However, the results for 2-propanol for the CAE in the informal sector showed a low detection frequency and relatively low concentrations of 2-propanol, as reported by Keretetse et al. [14]. In contrast, the omission of the 2-propanol did not affect the comparison of the SAE and CAE regimes within the formal sector. Another limitation of our study is the time gap between the self-assessment in the informal sector and the other sampling campaigns. The samples of the SAE in the informal sector were analyzed separately from the others. They were affected by decisions on the particular VOCs to include, as well as the slightly different LoDs. We tried to compensate for this by deploying the same LoD/LoQ for all sampling results, but this is arbitrary. Another drawback of the study is the lack of contextual information relevant to understanding the exposure results. As demonstrated previously [14], the number of clients, type of nail treatment, and bystander operations are important determinants of exposure. The poor completion of time/activity sheets, especially in the informal sector, seems an inherent drawback of self-assessment studies and does not generate valuable information to understand the exposure process. Passive sampling, requiring offline analysis, still needs interpretation and feedback from researchers. This limits the use of this method as a citizen science tool due to the lack of information regarding an association between exposure and participants’ behavior. In contrast, direct reading devices or wearable low-cost sensors could provide this type of information that could have an impact on the wearers’ behavior, thus reducing exposure [21], and they may be more cost-effective than traditional expert-supervised active sampling devices [22,23]. A technical limitation regarding diffusion samplers is that the badge or tube should be placed on the participant’s lapel without any obstructions, and this should be performed for a sufficient time. Since the diffusion rate is determined by concentration differences, a recently opened passive sampler that is not worn, or that is worn for a short period, will generate biased results. In our study, we used a cutoff for a sampling duration of three hours, but the results might still be inaccurate. Future SAE and citizen science studies will benefit from the further development of various wearable low-cost sensors that provide not only estimates of pollutants’ concentrations, but also complementary information to evaluate the results from the perspective of exposure assessment [24].

Despite the limitations, the results of the study demonstrate that, especially in cases with exposure scenarios where there are large concentration variances, an increase in the number of samples provides a better estimate of the central tendency, even though the SAE regime may generate ‘low-quality’ data compared to the CAE. Our findings support the conclusion by Nicas et al., (1991) that the analytical variability, or the quality of the measurement, contributes minimally to the total measurement variability, especially in scenarios with GSDs > 2 [4]. 

## 5. Conclusions

Self-assessment of exposure in occupational settings, including informal workplaces, seems to be a feasible and economical method to contribute to the expansion of exposure data, which enables a more reliable interpretation of exposure variation. In principle, SAE can be a citizen science tool in occupational health; however, passive sampling, requiring analytical costs and experts’ feedback, is not the ideal method to assess exposure. However, exposure science approaches and expertise to identify the exposure pathway and determinants of exposure are still needed to develop effective exposure control strategies.

## Figures and Tables

**Figure 1 ijerph-20-05459-f001:**
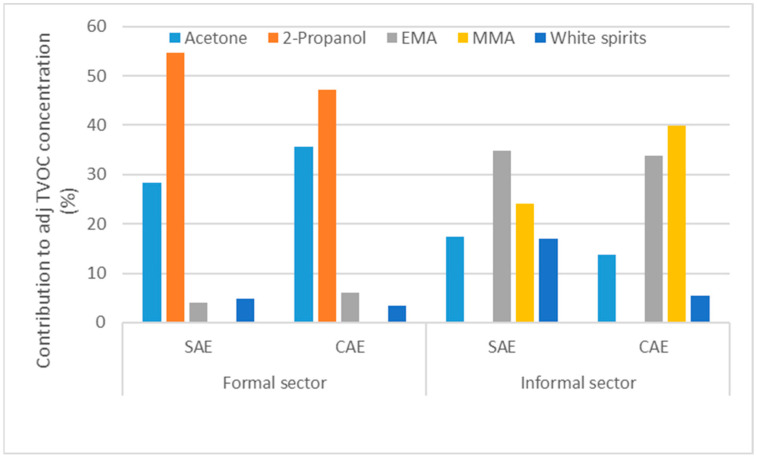
Major contributors to the adjusted TVOC-concentration (%) for both sectors and assessment regimes: SAE: self-assessment of exposure, CAE: controlled assessment of exposure. EMA: ethyl-methacrylate; MMA: methyl-methacrylate.

**Figure 2 ijerph-20-05459-f002:**
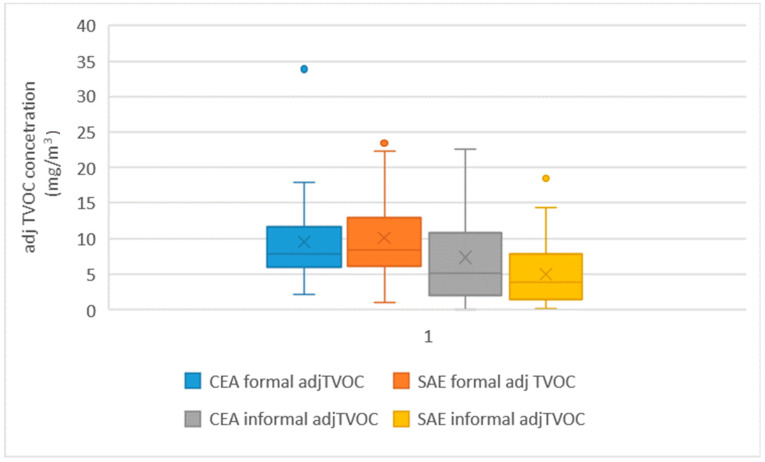
Box-plots of adjusted TVOC-concentrations (mg/m^3^) for both exposure assessment regimes. The dots represent the outliers.

**Table 1 ijerph-20-05459-t001:** Summary of VOCs analyzed and identified (>LoD) during SAE and CEA sampling campaigns.

	Formal				Informal			
VOC	SAE (*n* = 30)		CAE (*n* = 29)		SAE (*n* = 30)		CAE (*n* = 29)	
>LoQ	*n*	%	*n*	%	*n*	%	*n*	%
Acetone	28	93	29	100	29	97	29	100
Benzene	5	17	x	17	9	31	n.a.	n.a.
1-Butanol	n.a.	n.a	n.a	n.a	n.a.	n.a.	11	38
n-Butyl acetate	29	97	29	100	3	10	1	3
Chloroform	0	0	0	0	7	23	9	31
Ethyl acetate	29	97	29	100	29	97	24	83
Ethyl benzene	3	10	2	7	3	10	0	0
Ethanol	30	100	29	100	25	83	28	97
Ethyl methacrylate	30	100	26	90	28	93	24	83
d-limonene	27	90	28	97	11	37	13	45
Methyl ethyl ketone	7	23	3	10	n.a.	n.a.	1	3
Methyl-methacrylate	0	0	1	3	27	90	25	86
Methyl-isobutyl ketone	0	0	0	0	n.a.	n.a.	0	0
Perchloroethylene	1	3	n.a.	n.a.	1	3	n.a	n.a.
1-propanol	1	3	4	14	10	30	6	20
2-propanol	30	100	29	100	n.a.	n.a.	7	24
Propyl acetate	22	73	18	62	n.a.	n.a.	3	10
Toluene	29	97	28	97	18	60	26	90
Trichloroethylene	0	0	0	0	n.a.	n.a.	0	0
White spirits	25	83	24	83	26	87	27	93
Xylene	1	3	3	10	8	27	10	34

n.a.: not analyzed.

**Table 2 ijerph-20-05459-t002:** Summary statistics of the VOC shifts in average concentrations (mg/m^3^), which were selected for further analysis.

	Formal						Informal					
VOC	SAE (*n* = 30)			CAE (*n* = 29)			SAE (*n* = 27)			CAE (*n* = 29)		
	GM	GSD	IQR	GM	GSD	IQR	GM	GSD	IQR	GM	GSD	IQR
Acetone	65.7	6.63	54.4–169	108	2.49	64.7–162	10.9	8.67	3.3–56.7	24.3	5.11	11.3–78.7
n-Butyl acetate	0.59	2.11	0.33–0.87	0.62	1.97	0.41–0.96	n.a.	n.a.	n.a.	n.a.	n.a.	n.a.
Ethyl acetate	1.73	2.03	1.15–2.78	2.01	1.72	1.46–3.13	0.102	6.41	0.027–0.393	0.21	5.4	0.08–0.58
Ethanol	3.39	1.69	2.54–5.31	3.88	1.58	3.16–5.01	1.72	5.65	0.77–4.41	2.34	4.41	1.19–5.5
Ethyl methacrylate	1.49	4.33	0.62–3.4	1.56	6.94	0.39–5.67	6.24	8.18	2.08–29.3	3.57	14.9	0.35–32.8
d-limonene	0.19	2.79	0.12–0.28	0.14	2.63	0.09–0.21	0.0298	2.55	0.02–0.06	0.03	7.06	0.005–0.08
Methyl-methacrylate	n.a.	n.a.	n.a.	n.a.	n.a.	n.a.	1.89	18.3	0.135–23.7	8.06	9.48	2.41–54.5
2-propanol	46.7	2.21	28–91.5	43.7	1.72	33.8–58	n.a.	n.a.	n.a	n.a.	n.a.	n.a.
Propyl acetate	0.0469	2.32	0.03–0.07	0.0439	2.17	0.03–0.07	n.a.	n.a.	n.a	n.a.	n.a.	n.a.
Toluene	0.06	2.34	0.03–0.14	0.04	1.78	0.03–0.06	0.0372	2.01	0.0218–0.061	0.04	1.54	0.03–0.04
White spirits	0.94	5.96	0.2–3.41	0.45	4.76	0.1–1.19	1.02	5.11	0.27–3.39	0.9	2.46	0.49–1.67
Average		3.24			2.78			7.11			6.29	

n.a.: not applicable.

**Table 3 ijerph-20-05459-t003:** Restricted maximum likelihood estimates of the geometric mean (GM) exposure to adj TVOC in the formal and informal sector, separately and averaged, over the level of assessment regime (N_formal_ = 59, N_informal_ = 56). CI, confidence interval. The asterisk indicates a statistical significance difference.

Category	*n*	GM (mg/m^3^)		
		Estimate	Lower 95% CI	Upper 95% CI
Formal total	59	8.3 *	6.0	11.6
Informal total	56	3.6 *	2.4	5.3
Formal CAE	29	8.53	5.32	13.66
Formal SAE	30	8.17	5.74	11.64
Informal CAE	29	4.27	2.65	6.88
Informal SAE	27	3.08	2.07	4.59

* *p* = 0.011.

## Data Availability

The data are available upon request to the corresponding author.

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
