# Peer review of "Quantitative Self-Assessment of Exposure to Solvents among Formal and Informal Nail Technicians in Johannesburg, South Africa"

_ijerph, 2023, doi:10.3390/ijerph20085459_

Round 1
Reviewer 1 Report
This study is interesting. However, we would like to know the following specifications:
- In the first paragraph of Materials and methods, please clearly define the design and type of study.
- What is the scientific or knowledge contribution made by this study?
- How did you calculate the sample?
- In cases where the mean was calculated, please provide the Standard Deviation or Standard Error, as appropriate.
- In the case of the sample, please change the "N" to "n". This applies to the text in general, tables, and, where appropriate, figures.
- Provide the values of p.
- Could a risk assessment be carried out? Could OR values be calculated?
Author Response
Thank you for your comments. Please find our response in the file attached.

Reviewer 2 Report
The article is interesting and nicely written and it brings some useful information. I think that the manuscript has a potential to be published in the journal. However, I believe that the manuscript can be improved further. Thus I recommend the revising of the article based on the main points mentioned below.
- The novelty of the study is not properly highlighted in the Introduction section.
- In Abstract, line 12, please replace “of an SAE” with “of a SAE”.
- In all manuscript there are places where you let 2 spaces between the two words, so delete them (line 34: ”sciences [1,2]”; line 49: “later explored”; line 78: “polish, gel and”; line 157: “. The”; line190: “, 11”; line 194: “. Field”; line 199: “SAE in”; line 216: “sector,”; line 221:”in the” and “for all”; line 225: “is similar”; line 251: “was determined”; line 289: “differences in”; line 313: “study is”; line 331: “sampler that”; line 335: “where there”.
- Lines 67, 106, 119, 138: please delete the points after the subtitles.
- Line 82: replace “system.I In” with “system. In”.
- Line 125: Replace “μl and split flow of 20 ml/min” with “μL and split flow of 20 mL/min”.
- Lines 136, 137: Replace “ug/ml” with “μg/mL”.
- Please number the equation from line 170, page 4.
- Put in Italic font the terms from the equation (from lines 171-175).
- Line 175: please replace “denote” with “denotes”.
- Line 207: replace “na: not applicable” with “n.a.: not applicable”.
- Lines 215, 221: put the point at the end of the phrases.
- Line 218: replace “respecdtively” with “respectively”.
- Line 218: please revise “where the order swopped order during the CAE regime”, because the word “order” is repeating.
- Line: 267: delete one point from the end of the phrase.
- Line: 268: replace “detetected” with “detected”.
- Lines 281-299: the two paragraphs should be better placed in Introduction section.
- Please delete the references from the line 423 to 475, because you have put them twice.
Author Response
Thank you for your comments. Please find our comments in the file attached.

Reviewer 3 Report
The article is very original, relevant for the field, presented in a well-structured and organized manner, and provides very crucial information about the subject matter.
Very good background information which justifies the importance of the study, however, the authors need to state the research question more clearly and/or formulate their hypothesis (es) accordingly. Some of this information has already been included in the abstract but need to be in the introduction too.
Under Materials and Method, how did you determine which nail salons to include? Any sampling method used to select the salons? How did you determine which salon in Braamfontein area and in northern region of Johannesburg to include or exclude? Need more information on the inclusion and exclusion criteria used to determine eligibility.
The reported limitations of the study made no suggestion(s) for future work based on those outlined limitations.
Many of the references, eleven, are more than five years since published. May consider using more current literary sources as much as possible.
Author Response
Thank you for the comments. Please find our response in the attached file.

Reviewer 4 Report
Dear Authors
This is an interesting piece of work adopting the citizen science approach. To help improve the work quality consider this few comments
i. as VOCs respond to temperature, was this taken into account at any point as i noted sampling was done over 3 working days which present fluctuating temp as the da progresses.
ii. The technicians data was vaguely reported at start of the result and does not provided further insight to how this possibly influenced the SAE data or other results discussed
iii. Just a bit of clarity, it was reported that 20 VOCs were analysed however only 15VOCs were done for SAE, was it that the reported n.a. result was below LOQ or this was deliberately ignored as one will tend to assume the GCMS would have the ability to analyse all the reported VOCs.
iv. A bit of housekeeping will b required to improve few typo dotted in the manuscript
Best wishes
Author Response
Thank you for your comments. Please find our response in the attached file.

Round 2
Reviewer 1 Report
Thank you for responding to our comments. Please, include the information about the sample size calculation in the text of the manuscript. Thank you so much.
Author Response
see file attached-

Reviewer 2 Report
The authors have made all the corrections I suggested. Consequently, I recommend publishing the article in this form.
Author Response
thank you
Reviewer 3 Report
The authors have done a great job with the revision of this manuscript.
Author Response
thank you
Reviewer 4 Report
Many thanks for taking time out to improve the paper quality.
I will progress to recommend it for acceptance in its present format.
Author Response
thank you